# Kinetic Study of 17α-Estradiol Mechanism during Rat Sperm Capacitation

**DOI:** 10.3390/molecules27134092

**Published:** 2022-06-25

**Authors:** Tereza Bosakova, Antonin Tockstein, Zuzana Bosakova, Katerina Komrskova

**Affiliations:** 1Department of Analytical Chemistry, Faculty of Science, Charles University, Albertov 2030, Prague 2, 128 43 Prague, Czech Republic; terezabosakova@seznam.cz (T.B.); atockstein@seznam.cz (A.T.); 2Laboratory of Reproductive Biology, Institute of Biotechnology, Czech Academy of Sciences, BIOCEV, Prumyslova 595, 252 50 Vestec, Czech Republic; 3Department of Zoology, Faculty of Science, Charles University, Vinicna 7, Prague 2, 128 44 Prague, Czech Republic

**Keywords:** 17α-estradiol, estrogen receptors, rat, sperm, capacitation, HPLC MS/MS, kinetics

## Abstract

17α-Estradiol (αE2) is a natural diastereoisomer of 17β-estradiol (E2). It is well known that αE2 can bind to estrogen receptors. However, its biological activity is less than that of E2 and is species and tissue specific. The goal of our study was to propose the mechanism of αE2 hormonal response in rat sperm during their capacitation in vitro and compare it with a previously studied mouse model. Concentration changes in externally added αE2 during capacitation of rat sperm were monitored by the high-performance liquid chromatographic method with tandem mass spectrometric detection (HPLC-MS/MS). The calculated values of relative concentrations *B*_t_ were subjected to kinetic analysis. The findings indicated that αE2 in rat sperm did not trigger autocatalytic reaction, in contrast to the mouse sperm, and that the initiation of the hormone penetration through the sperm plasma membrane was substantially faster in rats.

## 1. Introduction

17α-estradiol (αE2) is a natural diastereoisomer of 17β-estradiol (E2) that can bind to estrogen receptors (ER), but its biological activity is substantially lower than the activity of E2 and it is species and tissue specific [1,2]. Its key role in the regeneration of injured brain tissue and its biological activity in relation to some tumor cells, such as testicular germ cell tumor (TGCT) has been described [2,3]. It has also been found that αE2, similarly to E2, is capable of bonding to estrogen receptors ERα and ERβ present in the plasma membrane of sperm, albeit with much lower affinity [4]. Recently, it has been shown that αE2 increases the life expectancy of mouse males and alleviates their metabolic and inflammatory disfunction connected with age without leading to feminization. αE2 improves sarcopenia associated with age and improves physical function at late mouse age, but only in males, and not in females or castrated males [5]. In relation to the fact that the increased presence of αE2 affects mouse males in various ways and does not affect females or castrated males [5,6], our previous research also investigated the effect of this hormone on mouse sperm during their capacitation [7].

Capacitation is a set of biochemical and physiological processes that sperm undergoes after ejaculation in the female reproductive tract and only capacitated sperm can fertilize an egg. Capacitation is also possible to stimulate in vitro under defined conditions, which are similar to the physiological environment [8,9].

Our previous study monitored concentration changes in αE2 during time-limited capacitation of mouse sperm in vitro in order to determine whether αE2 can bind to estrogen receptors in the sperm cytoplasm (cER). High performance liquid chromatography with tandem mass spectrometric detection (HPLC-MS/MS) was utilized during the capacitation to monitor the concentration of free αE2, unbound to sperm, and the resultant values of the relative concentrations underwent kinetic analysis. On this basis, a kinetic model was proposed, describing the accumulation of αE2 in the phospholipid bilayer of the plasma membrane under increasing tension such that when it attains critical value, it leads to penetration of the hormone through the membrane into the cytoplasm. αE2 reacts rapidly with cER in the cytoplasm to form an unstable adduct, which decomposes to the original components [7].

As described above, the biological activity of αE2 is tissue and species specific. It has been reported that αE2 regulates the expression of mRNA in tumor cells of the rat hypophyses, sensitive to estrogen. However, this induction is not mediated in the classical way by the estrogen receptor [10]. It is relevant to our study on sperm that it has been found that the part of the brain that affects spatial memory is rich in receptors that are sensitive to steroid hormones. After the estradiol administration, ovariectomized rats exhibited better performance and lower doses of hormone were sufficient for αE2 than for E2. This contrasts with the estrogen responses, which are generally more sensitive to the isomer of E2 [11,12]. The effect of αE2 indicates that hormone substitution therapy using this steroid could be clinically useful in alleviating the consequences of low endogenic production of estrogen on the development and progression of neurodegenerative disorders [13].

In relation to the above, experiments monitoring αE2 and its effect on the in vitro time-limited capacitation of rat sperm were performed. The HPLC-MS/MS method was utilized during capacitation to monitor the concentration of free, sperm-unbound αE2 and the obtained values of the relative concentrations underwent kinetic analysis. On this basis, a kinetic model was proposed and compared to the results previously published on mouse sperm [7].

## 2. Results

The concentration changes of αE2 unbound in rat sperm were measured according to the published methodology [7]. The HPLC-MS/MS method, working in reverse separation mode with a Kinetex EVO C18 column, was found to be useful and sufficiently sensitive for this purpose. The optimized separation and detection system was taken in its entirety from the work dealing with quantification of αE2 during the capacitation of mouse sperm [7]. The chromatogram of E2-d_3_ as an internal standard (IS) and αE2 as an analyte in capacitating M2 medium, measured under the optimized separation and detection conditions, is illustrated in Figure 1. The optimized conditions provided symmetrical peaks with sufficient separation efficiency. Due to the complex capacitating medium, the eluate was fed to the MS detector only from 2.5 to 5 min.

In the actual biological experiment, the dose concentration of αE2 was 200 μg/L, as in the previous study [7]. The experimentally obtained concentrations of free, sperm-unbound αE2 (*C*_t_), measured at the individual capacitation times are shown in Figure 2.

Concentration values of the samples (black points in Figure 2) were used to determine the relative concentration (*B*_t_), defined as *B*_t_ = *C*_t_/*C*_0_, where *C*_t_ is the concentration obtained for a given capacitation time and *C*_0_ is the concentration measured for capacitation time 0. The calculated values of *B*_t_ are given in Table 1. It is apparent from Table 1 that the relative concentration of biological samples initially decreases until a minimum is attained around 90 min of capacitation time and then begins to increase. Blank samples remain constant during the whole experiment.

In the previous study, kinetic models for the effect of hormones E2 [14], EE2 [15], and αE2 [7] on the capacitation of mouse sperm, were proposed together with the relevant rate constants, which were collectively characterized by autocatalytic formation of an adduct between the hormone and the estrogen receptor in the cytoplasm. The concept of an autocatalytic mechanism is based on a kinetic model describing the primary formation of an adduct in non-autocatalytic step, producing a signal for increased fluidity of the plasma membrane, which increases its permeability for the hormone. Therefore, a greater amount of extracellular hormone enters the cytoplasm by this autocatalytic step. However, the adduct is unstable and subsequently decomposes to an increasing concentration of free, in adduct unbound hormone. It is important for newly obtained data with rat sperm present previously publish kinetic scheme that was proposed for the interaction of αE2 with mouse sperm (see Figure 3).

This kinetic model was also adopted for the case of interaction of αE2 with rat sperm. However, in the rat, the course of *B*_t_ values does not appear to indicate any autocatalysis. The slope of the tangent to the hypothetical curve, obtained by interpolating experimental points *B*_t_, does not decrease in its initial region to zero with time, so the overall rate constant *K*_2_ corresponding to the autocatalytic step should be zero. Nevertheless, the obtained *B*_t_ values were subjected to kinetic analysis and three variables were selected for the mathematical description of the processes:(i)the concentration of αE2 hormone inside the cytoplasm, designated in the equations as *E*_i_;(ii)the concentration of the adduct of αE2 hormone with the estrogen receptor in the cytoplasm, designated as ER;(iii)the concentration of the inactive estrogen receptor in the cytoplasm, designated as *R*´.

Their changes during the time variation of the capacitation can be described by Equations (1)–(3) (taken from the work on the interaction of αE2 with mouse sperm [7]), where *E* is the concentration of extracellular αE2, *R* is the concentration of the estrogen receptor in the cytoplasm and (*ER*) is the concentration of the adduct:(1)$$\frac{-dE}{dt}=E\times R\left(K_{1}+K_{2}\times \left(1-\varepsilon \right)\right)$$
(2)$$\frac{-dR}{dt}=E\times R\left(K_{1}+K_{2}\times \left(1-\varepsilon \right)\right)+K_{4}\times E_{i}\times R-K_{3}\times \left(ER\right)$$
(3)$$\frac{d\left(ER\right)}{dt}=E\times R\left(K_{1}+K_{2}\times \left(1-\varepsilon \right)\right)+K_{4}\times E_{i}\times R-\left(K_{3}+K_{5}\right)\times \left(ER\right)$$

Equation (1) describes the decrease in hormone concentration (*E*) through the non-autocatalytic (*K*_1_) and autocatalytic (*K*_2_) reactions with receptor (*R*). Equation (2) describes the decrease in the receptor concentration through its reaction with the external (*E*) and internal (*E*i) pool of the hormone and *via* reverse reaction (*K*_1,_ *K*_2_, and *K*_4_) and the increase through decomposition of the adduct (*K*_3_). Equation (3) describes the increase in the adduct concentration through the non-autocatalytic and autocatalytic reactions and reverse reaction and the decrease through the equilibrium (*K*_3_) and pseudo-equilibrium (*K*_5_) reactions.

It is sufficient to express receptor concentration *R* for the balance conditions:*R*_0_ = *R* + *ER* + *R*´ for overall receptor concentration *R*_0_;*R* = *R*_0_ − *ER* − *R*´, where *ER* and *R*´ are variables as was mentioned above;*E*_0_/*R*_0_ = *n*, where *E*_0_ is overall hormone concentration and *n* is molar ratio.

We then factor out *R*_0_ and *R* = *R*_0_(1– *n*α − β), where

E_0_/*R*_0_ = *n*,→*E*/*E*_0_ = ε,→*ER*/*E*_0_ = α,→*R*´/*R*_0_ = β,→*t R*_0_ = τ (generalized time).

These relative concentrations were introduced into the kinetic equations, following from the overall kinetic scheme (if *K*_2_ = 0), yielding a set of 3 differential Equations (4)–(6) for ε, α and β.
(4)$$\frac{-d\varepsilon }{d\tau }=K_{1}\varepsilon \left(1-n\alpha -\beta \right)\;$$
(5)$$\frac{-d\alpha }{d\tau }=K_{1}\varepsilon \left(1-n\alpha -\beta \right)+K_{4}\left(1-\varepsilon -\alpha \right)\left(1-n\alpha -\beta \right)-K_{3}\alpha$$
(6)$$\frac{d\beta }{d\tau }=K_{5}n\alpha$$

This set of differential equations was solved by the Runge-Kutta method. To ensure the best fit of the resultant *B*(*t*) curve to the experimental points, the constants given in Table 2 were selected.

The value of constant *K*_2_, related to the autocatalytic formation of an unstable adduct, must equal zero. If the autocatalysis relates to an increase of the plasma membrane fluidity (as was true for E2 [14], EE2 [15] and αE2 in the interaction with mouse sperm [7]), then the mechanism without autocatalysis should mean that the resulting adduct ER does not affect the membrane fluidity and thus the amount of hormone transported across the membrane into the cytoplasm. If these constants are entered into the kinetic scheme (see Figure 3), it would result in the strongest process of the reverse formation of adduct ER (constant *K*_4_). The way the value of the molar ratio *n* affects the shape of the theoretical *B*(*t*) curves can be seen in Figure 4, from which it clearly follows that the most suitable value of the molar ratio *n* is 0.5 (red curve).

For comparison, Figure 5 depicts the shape of the *B(t)* curves calculated with the optimized kinetic parameters, together with the experimentally obtained points for the interaction of αE2 with (A) mouse sperm [7] and (B) rat sperm for a dose concentration of 200 μg/L.

## 3. Discussion

It can be seen from Figure 5 that the shape of the *B(t)* curves differs greatly for mouse and rat sperm. For mouse sperm (Figure 5 A), the flat part of curve (to 90 min capacitation) corresponds to difficulties encountered in penetration of hormone αE2 through the plasma membrane. The hormone accumulates between the phospholipid double layer of the plasma membrane with increasing tension and, when the amount of accumulated hormone attains critical value, the inner layer of the membrane releases the accumulated hormone which is internalized in the cytoplasm. The hormone within the cytoplasm, rapidly autocatalytically reacts with cER forming an unstable adduct (the rapidly descending part of curve), which subsequently decomposes (ascending part of curve). As the critical amount of αE2 compared to the overall amount of cER in the cytoplasm is very small, the value of molar ratio *n* is also very small (0.01). The difficulties encountered in the penetration of αE2 through the plasma membrane also correspond to the minimum on curve on the time axis, located around 150 min of capacitation (for details, see [7]).

For rat sperm (Figure 5B), a single almost symmetrical curve is obtained, with a minimum at a capacitation time of 90 min. In order to obtain a good fit of the experimentally obtained points with the calculated *B*(*t*) curve, it is necessary that the kinetic model does not contain an autocatalytic step (so that *K*_2_ = 0). The shape of the curve corresponds to continuous penetration of the hormone through the plasma membrane, with the formation of an unstable adduct of the hormone with cER (descending part of the curve) without signalization and a near mirror decomposition of the unstable adduct. The value of molar ratio *n* is also greater (*n* = 0.5) than that for mouse sperm (*n* = 0.01) [7]. Comparison of the initial values of rate constant *K*_1_ calculated for the reaction of the hormones with mouse sperm, EE2 (*K*_1_ = 0.03 [15]) and αE2 (*K*_1_ = 0.01 [7]), with the value of *K*_1_ obtained for rat sperm (*K*_1_ = 0.77), indicates that the reaction of αE2 with rat sperm is substantially faster, reflected in the steepness of the curve even at the beginning of the capacitation.

## 4. Materials and Methods

Relevant materials and methods were used in accordance with previously published kinetic studies [7,14,15], in order to guarantee comparison between the studied hormones.

### 4.1. Chemicals, Reagents and Animals

Commercial capacitating M2 culture media (M7167), deuterated β-estradiol-16,16,17-d_3_ (E2-d_3_) (purity 98%), acetonitrile (ACN) for LC-MS, Chromasolv (purity ≥ 99.9%), and phosphate buffered saline (PBS) tablets were purchased from Sigma-Aldrich^®^ (Steinheim, Germany). Formic acid (HCOOH) (LC-MS LiChropur, purity 97.5–98.5%), 17α-estradiol (purity 98%) and deionized water (for UHPLC-MS LiChrosolv) were provided by Merck (Darmstadt, Germany). Ethanol (purity ≥ 96% p.a.) was purchased from Lachner (Neratovice, Czech Republic). Paraffin oil was purchased from Carl Roth^®^ (Karlsruhe, Germany).

The experimental animals were laboratory inbred house rat strain Wistar (Velaz, Czech Republic). The males (*n* = 3) used for the experiments had reached sexual maturity and were 12 weeks old. The animals were housed in the animal facility of the Institute of Molecular Genetics of the Czech Academy of Science, Prague. Food and water were supplied *ad libitum*. All animal procedures and all experimental protocols were approved by the Animal Welfare Committee of the Czech Academy of Sciences (Animal Ethics Number 66866/2015-MZE-17214, 18 December 2015).

### 4.2. Instrumentation and Chromatographic Conditions

The Olympus epifluorescent microscope and Olympus CX 21 inverted-light microscope, purchased from Olympus, (Prague, Czech Republic) were used for sperm motility evaluation. The NB-203 incubator, purchased from N-BIOTEK (Bucheon, Korea), the BioTek laminar box and Telstar Bio-IIA incubator from N-BIOTEK (Bucheon, Korea) were used for the sperm cultivation in vitro.

The HPLC equipment from Agilent Technologies (Waldbronn, Germany) consisted of 1290 Infinity Series LC (quaternary pump, degasser, thermostatic autosampler, column oven). A Triple Quad LC/MS 6460 tandem mass spectrometer from Agilent Technologies (Waldbronn, Germany) with an electrospray ionization interface was used for the detection and quantitation of αE2. The signal and data were processed using the MassHunter Workstation Acquisition and MassHunter Qualitative Analysis Software from Agilent Technologies (Waldbronn, Germany).

All the instrumental MS-MS parameters were taken from [7,16]. For αE2 the transition *m*/*z* 255 → 159 (fragmentor voltage 130 V, collision energy 14 V) and for E2-d_3_ the transition *m/z* 258 → 159 (fragmentor voltage 100 V, collision energy 15 V) were used, respectively.

The separation system based on the publication [7] consisted of a Kinetex EVO column C18 (100 × 3.0 mm, 2.6 μm from Phenomenex (Torrance, CA, USA) and a mobile phase 50/50 (*v/v*) acetonitrile/water, both containing 0.1% formic acid. The eluate was fed to the MS detector from 2.5 to 5 min. The validation parameters were taken from the previously published paper, see [7] for details.

### 4.3. Sample Preparation and Evaluation

Petri dishes (35 mm), used for the capacitation in vitro, were obtained from Corning (Oneota, NY, USA). The working concentration of αE2 in the capacitating medium at a concentration 200 μg/L was prepared as described in [7]. Subsequently, 100 μL of the working solution was pipetted into the fertilization Petri dishes. The pipetted mixture in each Petri dish was covered with 1.5 mL of paraffin oil. The prepared Petri dishes were placed in the incubator and tempered for 60 min at a temperature of 37 °C and in the atmosphere with 5% CO_2_.

The conditions for rat sperm capacitation in vitro were the same [17] as for the mouse sperm and were performed according to the same laboratory protocol [18]. The spermatozoa which were recovered from the distal region of the *cauda epididymis* were placed in the fertilization Petri dishes with capacitating medium and paraffin oil and then placed for 10 min in the incubator for sperm release. Then the stock concentration of rat sperm was adjusted by PBS to 5 × 10^6^ sperm/mL. The capacitation progress was monitored by sperm motility, particularly changes in the tail beating pattern towards progressive motility, inspected under the microscopes.

The biological samples and blanks were prepared by the same procedure as described in [7]. Following 60 min of tempering, 5 µL of the stock sperm were added to 100 µL of the αE2 sample in capacitating medium (200 µg/L). For each capacitation time (0–180 min), 8 Petri dishes containing 105 µL of sample covered with 1.5 mL of paraffin oil were prepared. The dishes prepared in this way were incubated again under the same conditions for individual time periods (0, 30, 60, 90, 120, 150, and 180 min after the sperm were added), during which sperm capacitation took place. After these individual times, samples (only the capacitating medium without the paraffin oil) were pipetted out of all 8 Petri dishes into a single micro-test tube, which was centrifuged for 10 min at 12,000 rpm. In this way, the sperm (together with sperm-bound αE2) were separated from the solution and 600 μL of supernatant was collected for HPLC-MS/MS analysis of free, sperm-unbound αE2, and represented one sample for one capacitation time. Reference samples (blanks) without the addition of rat sperm were prepared simultaneously under the same experimental conditions. Samples were prepared in three parallel sets representing sperm collected from one male (*n*_sets_ = 3). Each sample was measured in five replicates by HPLC-MS/MS method for each capacitation time. The calculated average value (*n*_samples_ = 15) represents the mean concentration, obtained for one animal and for one sampling time. Totally, three animals were used (*n*_animals_ = 3) and error bars were calculated using standard deviation (*n*_animals_ = 3).

Prior to the actual HPLC-MS/MS analysis, 20 µL of IS (E2-d_3_) at a concentration of 250 µg/L in ethanol was added to each biological sample and a blank, to achieve a final concentration of 25 µg/L. No matrix effect was observed. The sample recovery was 95.7 %.

## 5. Conclusions

The results obtained for the interaction of hormone αE2 with rat sperm during the time-limited capacitation in vitro and the proposed kinetic model indicate a different mechanism of interaction and response than described for the action of hormones αE2 [7], E2 [14] and EE2 [15] in mouse sperm. Autocatalysis was not observed for αE2 in rat sperm, i.e., the formation of an unstable adduct does not determine greater penetrability of the plasma membrane for αE2 (zero value of *K*_2_). This is the major difference in the response triggered by αE2 during sperm capacitation compared to the kinetic models proposed for the interaction of the αE2, E2 and EE2 in capacitating mouse sperm. The calculated values of the rate constant for the primary formation of the adduct (*K*_1_) indicate that the reaction between capacitating rat sperm and αE2 hormone proceeds significantly faster with smoother formation of an unstable adduct than in mouse sperm. Based on the obtained results, it can be concluded that there could be possible hormone related species-specific signaling (which should be taken into consideration).

## Figures and Tables

**Figure 1 molecules-27-04092-f001:**
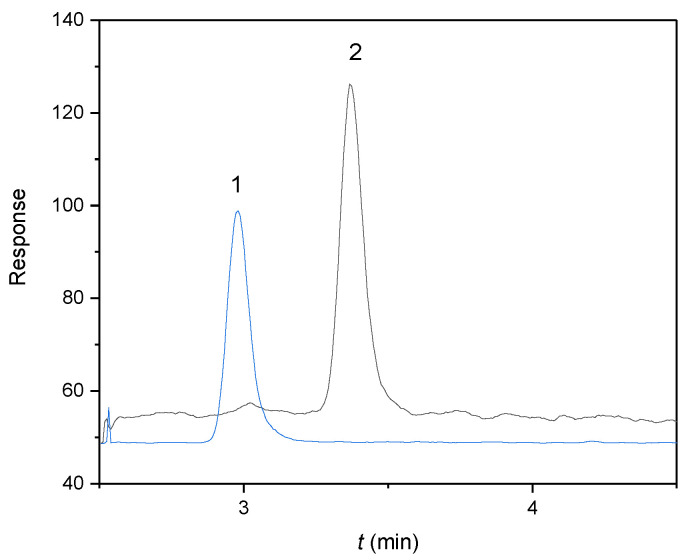
MRM chromatograms of (1) E2-d_3_ as an internal standard (IS) at a concentration of 25 μg/L (*t*_r_ = 3.0 min, MRM transition 258 → 159) and (2) αE2 as an analyte at a concentration of 20 μg/L (*t*_r_ = 3.4 min, MRM transition 255 → 159) in capacitating M2 medium. Experimental conditions: 50/50 (*v*/*v*) acetonitrile/water, both containing 0.1% formic acid, flow rate 0.3 mL/min, sample injected 10 μL.

**Figure 2 molecules-27-04092-f002:**
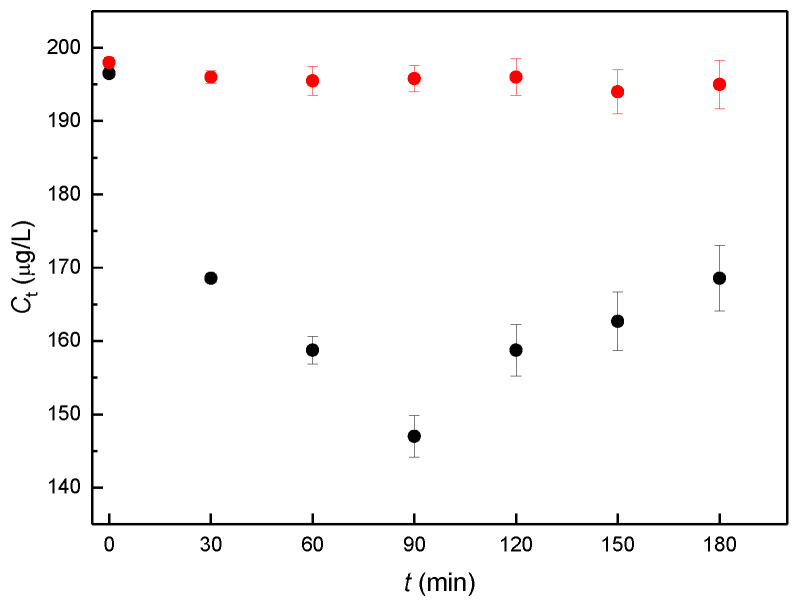
Concentration (*C_t_)* of free, sperm-unbound αE2 measured at the individual capacitation time using the optimized HPLC-MS/MS method. Each experiment was carried out for biological samples with added rat sperm (black circles) and for reference samples without sperm (blanks, red circles). Samples were prepared in three parallel sets representing sperm collected from one male. Three animals were used, and error bars were calculated using standard deviation (*n* = 3), for details see Section 4.3. Sample preparation and evaluation.

**Figure 3 molecules-27-04092-f003:**
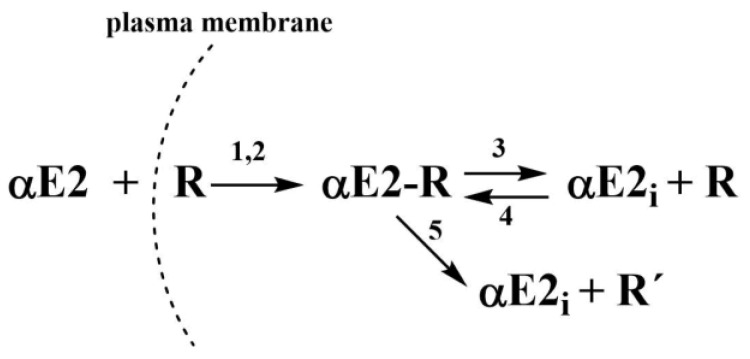
Kinetic scheme where αE2 is the extracellular hormone; R is the active sperm receptor; αE2-R is the adduct; αE2_i_ is the internal free hormone and R’ is the inactive sperm receptor. 1 and 2 represent the rate constants of the simultaneous non-autocatalytic and autocatalytic formation of the adduct αE2-R, 3 and 5 represent the rate constants of the adduct decomposition and 4 represents the rate constant of the reverse reaction (taken from [7]).

**Figure 4 molecules-27-04092-f004:**
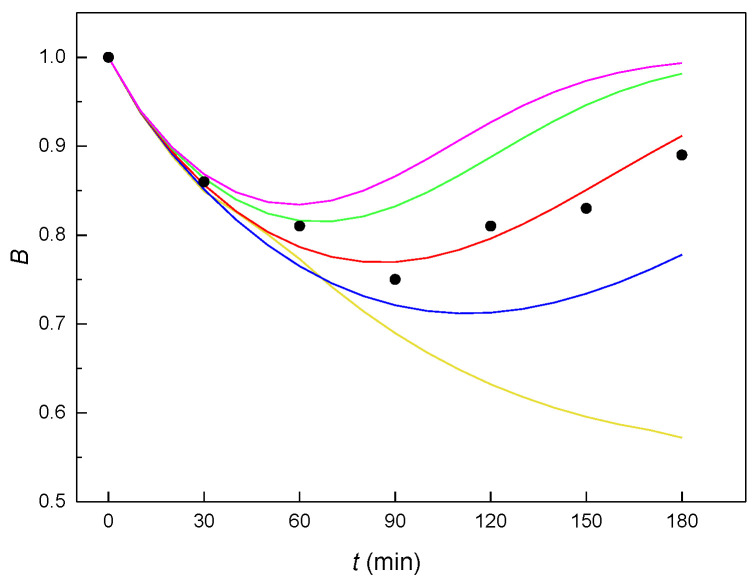
The course of the theoretical *B*(*t*) curves calculated with the optimized constants *K*_1_– *K*_5_ (see Table 2) for various values of the molar ratio *n* (yellow curve: *n* = 0.1; blue curve: *n* = 0.3; red curve: *n* = 0.5; green curve: *n* = 0.8; violet curve *n*: = 1.0; black circles—experimental points).

**Figure 5 molecules-27-04092-f005:**
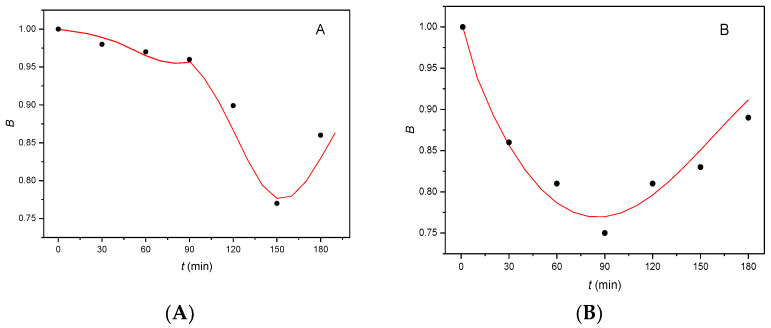
Comparison of the *B*(*t*) curves obtained for the interaction of hormone αE2 with (**A**) mouse sperm [7] and (**B**) rat sperm, dose concentration 200 µg/L; the experimentally obtained points are depicted as black circles.

**Table 1 molecules-27-04092-t001:** The relative concentration *B*_t_, obtained for the individual capacitation times, mean ± standard error of the mean (*n* = 3).

Capacitation Time (min)	0	30	60	90	120	150	180
*B* _t_	1.000 ± 0.003	0.860 ± 0.005	0.810 ± 0.008	0.750 ± 0.012	0.810 ± 0.015	0.830 ± 0.017	0.860 ± 0.019

**Table 2 molecules-27-04092-t002:** Calculated overall rate constants *K*_1_–*K*_5_ and molar ratio *n* for αE2 (concentration 200 µg/L).

*K* _1_	*K* _2_	*K* _3_	*K* _4_	*K* _5_	*n*
0.77	0.00	5.00	12.00	6.00	0.50

## Data Availability

Not applicable.

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
