# Peer review of "Kinetic Study of 17α-Estradiol Mechanism during Rat Sperm Capacitation"

_molecules, 2022, doi:10.3390/molecules27134092_

Round 1

Reviewer 1 Report

The paper describes in vitro model experiments for the comparison of the capacitation activity of stereoisomers of estradiol. The experiments are well-done and the evaluation modest and convincing. The results show substantial differnce between the activities of the 17-alpha and 17-beta stereoisomers and moreover these show that the effects are not general but species-specific. This aspect should be pointed out more explicitely in the paper.

The results show another general consequence, which should be mentioned in the paper. It has been found (in accordance also with earlier research) that both stereoisomers of estradiol do show biological activity, in the same sense, but with different mechanisms. This is an eclatant example of the fact, that the chirality in biomolecules should be better called as "biological chirality" and not "biological homochirality", where the latter would mean that oly one of the stereoisomers would show biological activity.

The background literature is suitable, however the recent book on chiral biomolecules should be added:

Pályi, G.: Biological Chirality. Elsevier-Academic Press, London, 2020.

An Editorial check of the English style and grammar appears to be useful.

Reviewer 2 Report

The authors conducted an interesting study on kinetic variations of 17α-oestradiol during in vitro capacitation of rat sperm cells. Despite being of interest in understanding the physiological changes occurring during sperm capacitation, the manuscript needs some improvement before its acceptance for publication.

First, the title does not agree with the content of the manuscript since in the present research authors only analysed the kinetic variations of 1717α-oestradiol in rats, but they did not in mouse. They only compare the results between both species in Figure 4. So, authors must provide a new title, according to the research conducted.

Lines 96-97. Please, provide a more appropriate title for Table 1.

Line 99. Please, add a Table foot indicating how the results are expressed and the sample size.

Line 194. Considering that the curve of Figure 4A belongs to a previous study, authors must cite that publication.

Line 234. Subsection does not content any information of the animals included in the study. So, authors must provide a new title for this subsection.

Lines 268-269. Please, indicate the number of animals included in the study and the age.

Line 299. Please, indicate the procedures used to measure the sperm motility and viability. Did the authors check the capacitation status of the sperm cells throughout the incubation time? If so, please provide a description of the procedure followed and show the results obtained in the Results section.

Author Response

Please see the atachment.

Round 2

Reviewer 2 Report

The authors have introduced all the modifications requested and the overall quality of the manuscript has been improved. Nevertheless, I am confident about the sample size since the study includes only three animals. Authors must justify from a statistical point of view that this sample size is enough to obtain robust results.
